



Natural Hazards
and Earth System
Sciences

# Modeling of *E. coli* distribution for hazard assessment of bathing waters affected by combined sewer overflows

**Luca Locatelli[1], Beniamino Russo[1,2], Alejandro Acero Oliete[2], Juan Carlos Sánchez Catalán[2], Eduardo Martínez-Gomariz[3], and Montse Martínez[1]**

[1]AQUATEC – Suez Advanced Solutions, Ps. Zona Franca 46-48, 08038, Barcelona, Spain
[2]Group of Hydraulic and Environmental Engineering (GIHA), Technical College of La Almunia (EUPLA),
University of Zaragoza, Mayor St. 5, 50100, Zaragoza, Spain
[3]Cetaqua, Water Technology Centre, Environment, Society and Economics Department,
Cornellà de Llobregat, 08940, Spain

**Correspondence:** Luca Locatelli (luca.locatelli@aquatec.es)

**Abstract.** Combined sewer overflows (CSOs) affect bathing water quality of receiving water bodies by bacterial pollution. The aim of this study is to assess the health hazard of bathing waters affected by CSOs. This is useful for bathing water managers, for risk assessment purposes, and for further impact and economic assessments. Pollutant hazard was evaluated based on two novel indicators proposed in this study: the mean duration of insufficient bathing water quality (1) over a period of time (i.e., several years) and (2) after single CSO/rain events. In particular, a novel correlation between the duration of seawater pollution and the event rainfall volume was developed. Pollutant hazard was assessed through a coupled urban drainage and seawater quality model that was developed, calibrated and validated based on local observations. Furthermore, hazard assessment was based on a novel statistical analysis of continuous simulations over a 9-year period using the coupled model. Finally, a validation of the estimated hazard is also shown. The health hazard was evaluated for the case study of Badalona (Spain) even though the methodology presented can be considered generally applicable to other urban areas and related receiving bathing water bodies. The case study presented is part of the EU-funded H2020 project BINGO (Bringing INnovation to OnGOing water management – a better future under climate change).

## 1 Introduction

Bathing water quality is regulated by the Bathing Water Directive (2006/7/EC) (BWD) and the corresponding transposition law within each EU nation. For instance, in Spain it is the Real Decreto 1341/2007. The BWD sets the guidelines for the bathing water monitoring and classification, the management, and the provision of information to the public. Short-term pollution events (having usual durations of less than 72 h) like the ones caused by combined sewer overflows (CSOs) lead to insufficient bathing water quality and require additional monitoring/sampling of bathing waters. Model simulations can be used to predict the pollutant plume spatial and temporal evolution in bathing water bodies; however, such tools are not widespread (Andersen et al., 2013). In the case of moderate and heavy rains, CSOs discharge high concentrations and loads of the bacteria *E. coli* (*Escherichia coli*) and intestinal enterococci (coming from wastewater and stormwater runoff) in the receiving water bodies where concentrations can exceed the bathing water quality standards. If bathing water quality is insufficient, then local authorities should inform end users, discourage bathing and collect water samples to monitor bacterial pollution. Generally, safe bathing can be reestablished after a collected water sample has shown acceptable bathing water quality.

In the field of risk management and considering a social-based risk approach, the risk can be assessed through the combination of the hazard likelihood and the vulnerability

of the system referring to the propensity of exposed elements – such as human beings, their livelihoods, and assets – to suffer adverse effects when impacted by hazard events (BINGO D4.1, 2016). In this framework, risk can be defined as the combination of hazard and vulnerability (including exposure, sensitivity and recovering capacity) according to the CE1 literature (Turner et al., 2003; Velasco et al., 2018). Donovan et al. (2008) and Viau et al. (2011) evaluated the risk of gastrointestinal disease associated with exposure of people to pathogens like *E. coli* and enterococci. In the former study, hazard was assessed by statistical analysis of observed bacterial concentrations during 6 d TS1 in a year that was considered representative, whereas in the latter one it was estimated by simple assumptions. Andersen et al. (2013) presented a coupled urban drainage and seawater quality model to quantify microbial risk during a swimming competition where lots of gastrointestinal illnesses occurred due to the presence of CSO in seawater. O'Flaherty et al. (2019) evaluated human exposure to antibiotic-resistant *Escherichia coli* through recreational water.

Several water quality models of receiving water bodies were developed to simulate spatial and temporal variations of bacterial concentrations originating from CSOs and other sources. These water quality models also include hydrodynamic models most of the time. Scroccaro et al. (2010) developed a 3D seawater quality model to simulate bacterial concentrations originating from wastewater treatment plant discharges. Jalliffier-Verne et al. (2016) and Passerat et al. (2011) developed river water quality models. Liu and Huang (2012) developed a 2D model of an estuary exposed to tides. Sokolova et al. (2013) and Thupaki et al. (2010) presented hydrodynamic 3D models of lakes to simulate *E. coli* based on pollutant discharges estimated from observations at affluent rivers and/or sewers. Also, coupled urban drainage and water quality models of receiving water bodies were developed to simulate spatial and temporal variations of bacterial concentrations for bathing water quality affected by CSOs (Andersen et al., 2013; De Marchis et al., 2013).

None of the studies presented above provided a methodology that combined simulated *E. coli* concentration with hazard criteria to evaluate the health hazard of bathing waters affected by CSOs that is the main aim of this study. Health hazard was evaluated based on two novel indicators proposed: the mean duration of insufficient bathing water quality (1) over a period of time (i.e., several years) and (2) after single CSO/rain events. In particular, a novel correlation between the duration of seawater pollution and the event rainfall volume is presented. This is useful for bathing water managers, for risk assessment purposes, and for further impact and economic assessments. For example, the presented correlation can be useful to water managers and regulators to predict how long a rainfall event is going to affect the bathing water quality and when the optimal time to collect bathing water samples is CE2. Also, it can be useful to estimate direct and indirect economic impacts of CSOs on

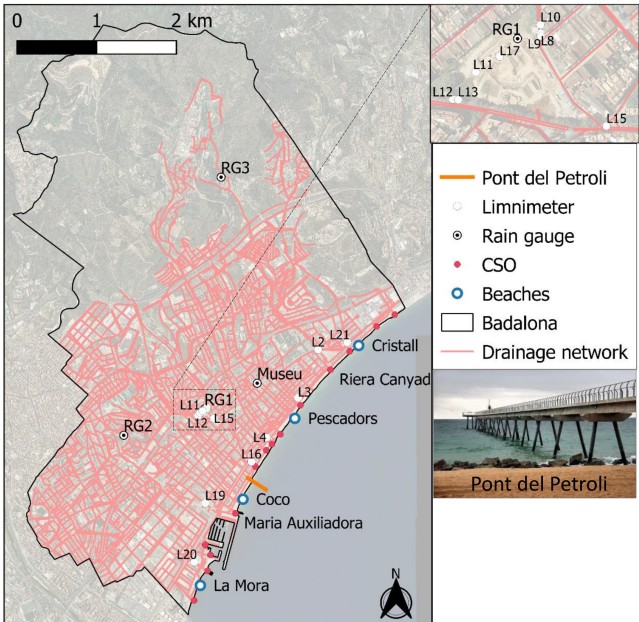

**Figure 1.** Plan view of Badalona together with the drainage network, the name of some of the beaches, the CSO points, the rain gauges, the limnimeters and the pedestrian bridge Pont del Petroli. Background image from © Google Maps.

coastal economies as was done in the BINGO project. *E. coli* concentration in the receiving water body was simulated by a coupled urban drainage and seawater quality model that was developed, calibrated and validated based on local observations. The health hazard was then quantified through the coupling of simulated *E. coli* concentrations and hazard criteria that were defined based on the BWD specifications. Furthermore, a novel statistical analysis of continuous simulations over a 9-year period using the coupled model is presented. Finally, a validation of the estimated hazard is also shown. The health hazard of bathing waters affected by CSOs was evaluated for the case study of Badalona (Spain).

## 2 Materials and methods

### 2.1 The case study

Figure 1 shows the case study area of Badalona (Spain). Badalona, the fourth largest city of the Catalonia region, is part of the metropolitan area of Barcelona, with an area of $21 \, \text{km}^2$, 215 000 inhabitants and high urbanization. It has approximately 5 km of sandy bathing beaches facing the Mediterranean Sea. Several CSO points discharge combined sewers along the beaches. Generally, rainfall events larger than a few millimeters cause CSOs, and during the bathing season bathing is usually forbidden during at least the 24 h following a CSO event.

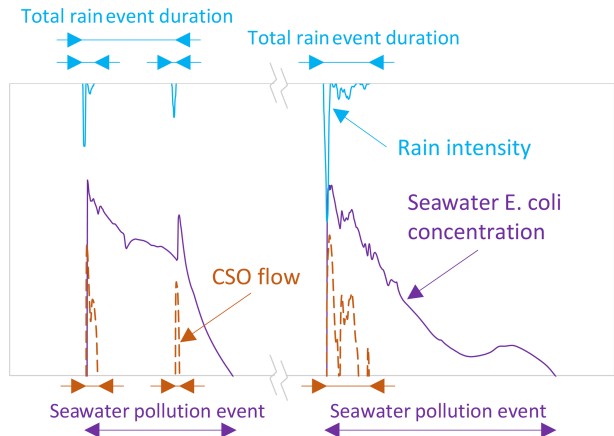

**Figure 2.** Definition of total rain event duration and seawater pollution event.

## 2.2 Definition of seawater pollution events

Figure 2 shows the definition of a total rain event duration and a seawater pollution event. Two different seawater pollution events are shown for an easier clarification of the definition adopted in this study. The figure shows three different rain events, each of them causing CSOs into the sea, and two different seawater pollution events. Seawater pollution events are defined as occurring when bacterial concentrations exceed the selected thresholds. A seawater pollution event can last up to a couple of days and can be generated by different CSO/rain events. Therefore, the definition of a total rain event duration is considered practical for this study considering also the different timescales of the different events involved. A similar definition was introduced in other urban water quality studies analyzing the performance of urban drainage structures such as detention ponds and basins on receiving water bodies (Sharma et al., 2016).

## 2.3 The data

This section provides an overview of the data collected for the case study. Figure 1 shows the location of the four rain gauges and 14 water level sensors that have been operating since 2011. These data were relevant for the calibration of the urban drainage model (Sect. 3.1). The rain gauges RG1, RG2 and RG3 were installed and have been operational since 2014, providing the time of each tipping of the 0.1 mm bucket capacity. The rain gauge at the Museu site was installed and has been operational since 2002, and open-access daily rainfall data are available.

Other sensors were also installed in 2015 (as part of the H2020 BINGO project) at the two CSO points of Maria Auxiliadora and Riera Canyadó (see Fig. 1): a turbidity and a temperature sensor upstream and a temperature and a water level sensor downstream of each of the two weirs. When a CSO occurs, both temperature sensors indicate approxi-

mately the same value, and the water level sensor is activated, so it is possible to detect the duration of the overflow and the frequency of occurrence of this type of event. These water level data were used for calibration of the simulated CSO hydrographs (obtained by the urban drainage model) at the two observed CSO points (Sect. 3.1). Also, two automatic 12-bottle samplers were installed to measure CSO turbidity, dissolved oxygen demand, suspended solids, and enterococci and *E. coli* concentrations at the two CSO structures. The measured *E. coli* concentrations at CSOs were used for the estimation of CSO concentrations used as inputs for the seawater quality model (Sect. 3.2). Turbidity, dissolved oxygen demand, suspended solids and enterococci concentrations were used for other purposes (see H2020 BINGO and LIFE EFFIDRAIN projects) out of the scope of this paper.

Figure 3 shows both the *E. coli* concentration and the CSO water level measured at the two monitored CSO points during the only two events registered: the 30 May 2017 event had 3–4 mm of rain in 3 h and the 24 March 2017 event had 55–67 mm in 12 h. During the latter event only data from one (Maria Auxiliadora) of the two monitored CSO structures were available. The water level sensors are located after the weirs, and they measure the CSO water levels. Instead, the bottle samplers for *E. coli* concentration measurements were located inside the CSO chamber. Generally, bottle sampling started (the first bottles were filled) before the onset of CSOs. During the CSO event of 30 May 2017, bottle sampling also continued after the end of the CSO. Instead, in the event of 24 March 2017 bottle sampling only covered the first approximately 40 min of the CSO event. The automatic bottle sampling frequency was set every 3 min for the first bottles up to 30 min for the last ones. Figure 3a shows four *E. coli* measurements during the CSO (CSO is identified by CSO water levels greater than zero) and Fig. 3b only two. These observations are between $5.7 \times 10^5$ and $1.0 \times 10^6$ CFU $(100 \, \text{mL})^{-1}$ (CFU: colony forming units). Figure 3c shows three *E. coli* measurements between $8.2 \times 10^4$ and $2.3 \times 10^5$ CFU $(100 \, \text{mL})^{-1}$ during the first 40 min of the 12 h long CSO. Further analysis of such observation was not done due to the sparse data.

Seawater quality data were measured by the laboratory technicians of the municipality of Badalona during five different field campaigns (in the period 2016–2018) that consisted of taking seawater samples after (sometimes also during) CSO events. Seawater samples were manually taken at three different points (1 – close to the shoreline, 2 – in the middle of the pedestrian bridge and 3 – at the most offshore point) along the approximately 200 m long pedestrian bridge Pont del Petroli (Fig. 1). The samples were taken once a day (normally between 09:00 and 14:00) for the few days following CSO events until bacterial concentrations recover to small values. The data measured were as follows: *E. coli* and intestinal enterococci concentrations, salinity, turbidity, and suspended solids. Seawater quality data are also available from the continuous water quality sampling campaigns

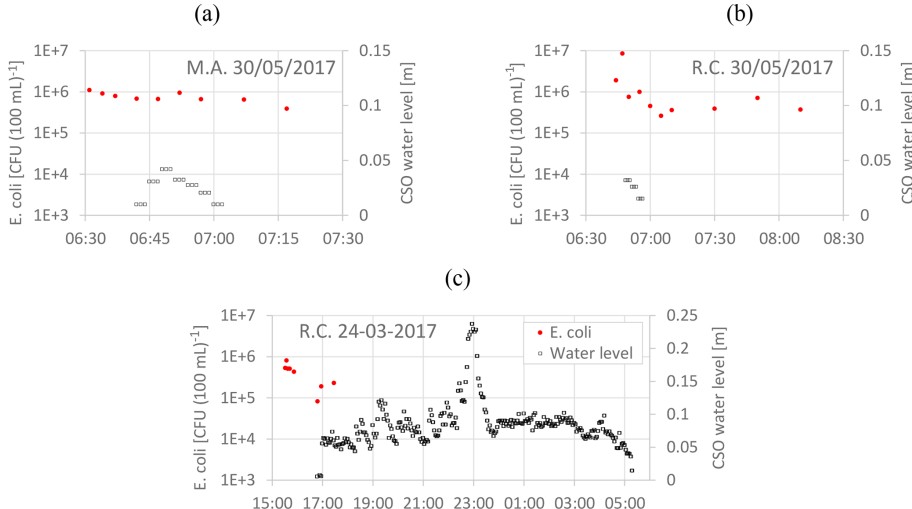

**Figure 3.** *E. coli* concentration and CSO water level measured at the two CSO structures of Riera Canyadó (R.C.) and Maria Auxiliadora (M.A.).

that are mandatory in order to classify the water quality at the different bathing locations. For instance, during recent years, more than 400 water samples were collected (and analyzed) at each of the four beaches shown in Fig. 1 with a sampling frequency of approximately once every couple of weeks. All the water quality indicators were obtained from laboratory analysis of the collected samples. Observed seawater *E. coli* and salinity concentrations were used for the calibration and validation of the seawater quality model (Sect. 3.2), and *E. coli* concentrations were also used for the validation of the hazard assessment (Sect. 3.3.1). Seawater turbidity, suspended solids and potential oxygen reduction data were not used in this study.

## 2.4 The model

A coupled urban drainage and seawater quality model was developed, calibrated and validated based on local observations. The urban drainage model is used to simulate CSO hydrographs at all the CSO points of Badalona. These hydrographs are used as inputs for the seawater quality model to simulate nearshore water quality. The two models are coupled in a sequential way; this means that first the urban drainage and then the seawater quality model are executed. This is acceptable as the physical processes occurring in the sea do not affect CSO hydrographs in Badalona.

### 2.4.1 The urban drainage model

The urban drainage model aims to simulate CSO hydrographs at all the CSO structures of Badalona that will then be used as inputs for the seawater quality model. The model simulates rainfall-runoff processes, domestic and industrial sewage water fluxes, and hydrodynamics in the drainage network over the whole area of Badalona.

The model was originally developed in MIKE URBAN (https://www.mikepoweredbydhi.com/, last access: 1 April 2020) for the 2012 drainage management plan (DMP) of Badalona. As part of this study, the model was imported into InfoWorks ICM 8.5 (https://www.innovyze.com/en-us, last access: 1 April 2020), updated to include the new pipes and one detention tank of approximately 30 000 m³ that were constructed during recent years and calibrated and validated with new data. Figure 1 shows the modeled drainage network. Overall, the model includes approximately 368 km of pipes, 11 338 manholes, 11 954 sub-catchments, 62 weirs, 4 sluice gates and 1 detention tank.

The sewer flows were simulated by the full 1D Saint-Venant equations. Rainfall-runoff processes were simulated for each sub-catchment using a single nonlinear reservoir with a routing coefficient that is a function of surface roughness, surface area, terrain slope and catchment width. Initial losses are generally small for both impervious urban areas and green areas ($\leq 1$ mm), and continuous losses for green areas are simulated using the Horton model. The area of Badalona was divided into 11 954 computational sub-catchments of different areas. The sub-catchments were obtained by GIS analysis of the digital terrain model (2 m × 2 m resolution) and have areas in the range of 0.01–1 ha in the urban areas and 1–100 ha in the upstream rural areas. Each sub-catchment includes the GIS-derived information of impervious areas and pervious areas that are used to apply either the impervious or the pervious rainfall-runoff model. Impervious areas do not have continuous hydrological losses, meaning that all the net rainfall (after initial losses) contributes to stormwater runoff.

## Calibration and validation of the urban drainage model

The urban drainage model was calibrated using a trial-and-error approach with the objective of minimizing the sum of all the root mean square errors (RMSEs) calculated for each of the observation points. The RMSE was calculated for the duration of each simulated event, usually an hour before the beginning of the rainfall and some hours after the end. Table 1 shows the three events selected for calibration and the one for validation. In addition, the different rainfall intensities, volumes and return periods evaluated based on local rainfall intensity–duration–frequency curves are shown for each event. It is noted that the selected calibration events have generally high intensities and volumes compared to the majority of the events that can cause CSOs (as mentioned above, events larger than a few millimeters usually cause CSOs in Badalona). These events were considered ideal for calibration and validation because of both (i) the significant observed water level variations in the drainage pipes and (ii) the overall quality and quantity of the collected rainfall and water level data. Rain data from the rain gauges were applied in the model using Thiessen polygons.

TS2 The calibrated Horton parameters were as follows: an initial infiltration capacity of $20\,\text{mm}\,\text{h}^{-1}$, a final infiltration capacity of $7\,\text{mm}\,\text{h}^{-1}$, a capacity decrease exponent of $0.043\,\text{h}^{-1}$ and a capacity increase exponent of $0.108\,\text{h}^{-1}$. The initial loss was not calibrated and it was calculated as *value/slope*$^{0.5}$, where *value* was set to 0.000071 and 0.00028 m for impervious and pervious surfaces respectively and *slope* was the average slope of the sub-catchment calculated with GIS. The calibrated Manning roughness coefficient of surface impervious areas was set to 0.013, and the coefficient of pipes was set to 0.012. The calibrated Manning roughness coefficients are in the lowest parts of the ranges proposed in similar urban drainage studies (Fraga et al., 2017; Locatelli et al., 2017; Russo et al., 2015) and more general hydrologic studies (Dingman, 2015; Henriksen et al., 2003).

Figure 4 shows two examples of simulated and observed water levels at two different locations during two different rain events. All the other graphs can be found in BINGO D3.3 (2019). Table 2 shows the computed RMSE and absolute maximum error (AME) (Bennett et al., 2013) for the calibration and validation events. The magnitude of the errors is similar to the ones reported in other urban drainage models (Russo et al., 2015). Overall, the model simulates the observed water levels in the sewer network in an acceptable way.

Finally, after the calibration and validation, a further manual adjustment of the simulated CSO flows at the two monitored CSO structures was performed using measurements of CSO water levels during three different CSO events from 2017 (67 mm on 24 March, 22 mm on 27 April and 4.2 mm on 30 May) and CSO structure geometry (weir crest level and width) of the two monitored CSO structures. The sim-

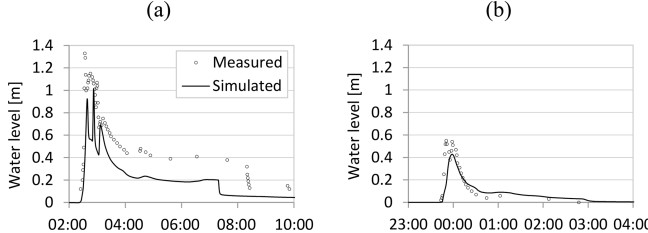

**Figure 4.** Example of simulated and measured water levels in the sewer pipes at the location BA15 during the rain event of 3 October 2015 **(a)** and at BA2 on 14 September 2016 **(b)**.

ulated crest level of these two weirs was manually adjusted by a few centimeters so that the simulated CSO water levels would better fit (visual judgment) the observed ones. It was verified that this further model adjustment did not affect the errors provided in Table 2. The crest level and width values of all the other CSO structures were obtained from the database of the network that came from the DMP of Badalona of 2012.

### 2.4.2 The seawater quality model

The seawater quality model aims at simulating nearshore (within few hundred meters from the shoreline) bacterial concentrations in the Badalona seawater during and after CSO events. The water quality model was developed for the area of Barcelona (Gutiérrez et al., 2010), and it has been operating since 2007 for real-time simulations of bathing water quality of the Barcelona beaches. This model was updated, calibrated and validated for Badalona as part of this study.

The seawater quality model was developed using the software MOHID by MARETEC (Marine and Environmental Technology Research Center) of the Instituto Superior Técnico (IST). The model simulates both the hydrodynamics of the sea in the coastal region and the pollutant transport resulting from CSOs.

Simulation of nearshore water quality and hydrodynamics during and after CSOs requires spatial discretization scales on the order of tens of meters, whereas other coastal hydrodynamic processes can occur at much larger scales. Therefore, three model domains are used to simulate hydrodynamic processes from the large regional scale to the local nearshore scale of Badalona. Figure 5 shows the three model levels. Level 3 represents city-scale processes, O(10 km); and is nested into Level 2, which represents subregional scale processes, O(50 km); and is further nested into Level 1, regional scale processes, O(200 km) TS3. The three levels continuously interact with each other during simulations. Level 1 covers an area of approximately $20\,000\,\text{km}^2$ with 6500 squared cells of approximately $1\,\text{km}^2$. At this domain, the hydrodynamic processes of astronomic tides and wind-induced waves and currents are simulated in 2D mode (barotropic). Level 2 covers an area of approximately 1000 TS4 $\text{km}^2$ with $54\,000$ rectangular cells of sides from 500 to 200 m (finer

**Table 1.** Events used for calibration and validation of the urban drainage model.

| Date event | $P$ (mm) cumulative rainfall RG1, RG2, RG3 | $I_{20\,min}$ (mm h$^{-1}$) maximum 20 min rainfall intensity ($T =$ return period) | $I_{5\,min}$ (mm h$^{-1}$) maximum 5 min rainfall intensity ($T =$ return period) | Event used for |
|---|---|---|---|---|
| 22 August 2014 | 18.6, 17.4, 26.0 | 42.6 ($T = 0.4$ years) | 74.4 ($T = 0.6$ years) | Calibration |
| 28–29 July 2014 | 46.5, 36.0, 2.8 | 56.4 ($T = 0.7$ years) | 91.2 ($T = 0.8$ years) | Calibration |
| 3 October 2015 | 33.4, 34.1, 26.0 | 81 ($T = 2.3$ years) | 122.4 ($T = 1.1$ years) | Calibration |
| 13–14 September 2016 | 30.2, 25.2, 20.2 | 64.5 ($T = 1.1$ years) | 142.8 ($T = 1.3$ years) | Validation |

**Table 2.** Root mean square error (RMSE) and absolute maximum error (AME) for the calibration and validation events.

| | Calibration | | | | | | Validation | |
|---|---|---|---|---|---|---|---|---|
| | 22 August 2014 | | 28 July 2014 | | 3 October 2015 | | 14 September 2016 | |
| Water level sensor | RMSE (m) | AME (m) | RMSE (m) | AME (m) | RMSE (m) | AME (m) | RMSE (m) | AME (m) |
| L2 | 0.066 | 0.062 | 0.065 | 0.028 | 0.200 | 0.215 | 0.181 | 0.102 |
| L4 | 0.045 | 0.033 | 0.049 | 0.218 | 0.150 | 1.422 | 0.057 | 0.022 |
| L8 | 0.008 | 0.075 | 0.005 | 0.032 | 0.011 | 0.055 | 0.021 | 0.027 |
| L9 | 0.157 | 0.666 | 0.000 | 0.322 | 0.006 | 0.733 | 0.001 | 0.321 |
| L10 | 0.103 | 0.725 | 0.098 | 0.501 | 0.087 | 0.472 | 0.101 | 0.128 |
| L11 | 0.212 | 0.627 | 0.222 | 1.150 | 0.495 | 2.390 | 0.244 | 1.333 |
| L12 | 0.123 | 0.019 | 0.167 | 0.340 | 0.295 | 0.790 | 0.189 | 0.169 |
| L13 | 0.094 | 0.315 | 0.094 | 0.565 | 0.713 | 0.907 | 0.238 | 0.631 |
| L15 | 0.236 | 1.036 | 0.195 | 0.656 | 0.322 | 0.315 | 0.183 | 0.791 |
| L16 | 0.264 | 0.537 | 0.032 | 2.065 | 0.850 | 1.828 | 0.147 | 0.357 |
| L19 | 0.131 | 0.214 | 0.099 | 0.617 | 0.209 | 0.686 | – | – |
| L20 | 0.346 | 0.387 | 0.148 | 0.190 | 0.333 | 0.059 | 0.096 | 0.159 |
| L21 | 0.056 | 0.290 | 0.082 | 0.222 | 0.099 | 0.190 | 0.181 | 0.300 |

cells close to the shoreline). Level 3 covers an area of approximately 50 km$^2$ with 117 528 rectangular cells of sides from 200 to 40 m (finer cells close to the shoreline). The vertical discretization of Levels 2 and 3 was defined with a sigma approach with thinner cells close to the sea surface and thicker ones close to the seabed. The percentages used to define the thickness of each of the eight vertical layers as a function of the water depth were as follows: 0.458, 0.227, 0.134, 0.079, 0.047, 0.028, 0.017 and 0.01 (the thinnest layer at the sea surface is 1 % of the water depth at that location). At Level 2 and Level 3 domains the processes of currents and waves, density, temperature and salinity variations; nearshore currents, generated by CSOs and river discharges; and advection and dispersion of *E. coli* from CSOs are simulated in baroclinic mode with a 3D mesh CE3.

CSOs are simulated in the seawater model as both water discharges and concentration inputs. Water discharge time series at the CSO points were obtained from the urban drainage model, and the input concentrations used for CSO discharges were assumed to be fixed (further details are given in the following section).

The hydrodynamic model solves the primitive continuity and momentum equations for the surface elevation and 3D velocity field for incompressible flows, in orthogonal horizontal coordinates and generic vertical coordinates, assuming hydrostatic equilibrium and Boussinesq approximations (http://wiki.mohid.com, last access: 23 April 2020). The selected turbulence model was the Smagorinsky model with default values. Wave height and period were computed as a function of wind speed, according to ADIOS model formulations (NOAA, 1994).

**Calibration and validation of the seawater quality model**

Three different events were used: two for calibration (one in January 2018 and one in September 2016) and one for validation (October 2016). These events were selected because they were the ones with the largest number of seawater *E. coli* and salinity measurements mainly due to both the relatively high *E. coli* concentrations observed and their slow recovering pollutographs. Three events may sound like a limited number for calibration and validation. This was because of the long computational time of the coupled model and because the data required for bathing water quality models are generally sparse and limited to some events. Similar models in the literature were also based on sparse data and few events

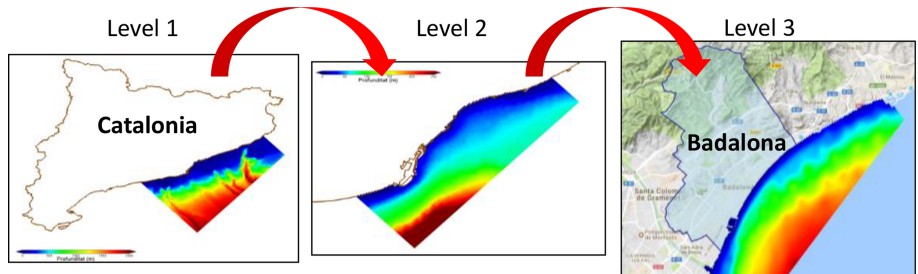

**Figure 5.** The three model domains of the seawater quality model. The colors represent the bathymetry (blue represents shallow and red represents deep). Background image from © Google Maps.

for calibration and validation: De Marchis et al. (2013) and Passerat et al. (2011) used a single event, and Andersen et al. (2013) used two events. The computational time of the seawater quality model was on the order of 2 h per each simulated day using an Intel® Core™ i5-6200U CPU 2.3 GHz processor.

Calibration was based on a trial-and-error approach trying both to optimize the visual fitting between observed and simulated values and to minimize the computed errors. Both *E. coli* concentrations and salinity were used in the calibration process. The two calibration parameters (wind drag coefficient and *E. coli* decay rate) were selected after a sensitivity analysis (BINGO D3.3, 2019). A fixed *E. coli* concentration of $1 \times 10^6$ CFU $(100 \, \text{mL})^{-1}$ was used as input for the CSO hydrographs. This is a significant influential parameter, and such a choice was justified after literature review and by the available observed data that were shown in Fig. 3. Different approaches were presented in the literature: Andersen et al. (2013) simulated CSO dilution using a drainage model with a fixed *E. coli* concentration for wastewater based on literature review and assuming clean stormwater runoff. De Marchis et al. (2013) used five events with river discharge and *E. coli* measurements to calibrate both water quantity and quality from the modeled sub-catchment. Jalliffier-Verne et al. (2016) estimated the CSO concentrations based on a fixed discharge per person multiplied by the number of people connected to the sewer network. Passerat et al. (2011) observed *E. coli* concentrations of $1.5 \times 10^6$ CFU $(100 \, \text{mL})^{-1}$ for a CSO where 89 % of the discharge was estimated to be from stormwater runoff. McCarthy et al. (2008) analyzed 56 wet rainfall events between 3.2 and 25 mm at four different catchments to estimate uncertainty and event mean concentrations of *E. coli*. Wind data for these events were obtained from Puertos del Estado (http://www.puertos.es, last access: 1 April 2020). In particular, a model reproduced observed wind over a selected cell (approximately 10 km long) that covered the area of Badalona, and such a wind speed was uniformly applied to the seawater quality model.

The calibrated wind drag coefficient was 0.0008. Generally, the higher the coefficient, the higher the seawater velocities and consequent pollutant advection and dispersion.

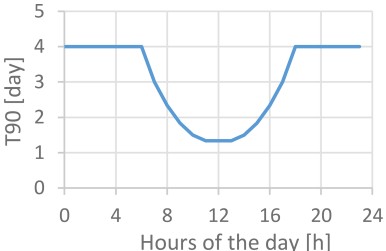

**Figure 6.** Calibrated *E. coli* decay rate.

The model MOHID allows either a user-defined fixed value that is suggested to be 0.0015 or the use of the function of Large and Pond (1981) to compute the wind drag coefficient as a function of the wind speed. The calibrated value is within the proposed ranges of Large and Pond (1981), and Sokolova et al. (2013) used 0.001255. Figure 6 shows the calibrated *E. coli* decay rate expressed as T90, defined as the time at which 90 % of the bacterial population is no longer detectable, meaning a one log reduction of the number of pathogens. T90 could be computed as a function of water temperature, salinity and solar radiation (Canteras et al., 1995; Sokolova et al., 2013). However, such formulations were tested and produced excessive decay rates for this case study; therefore the decay rate was assumed to have a daily pattern that was calibrated. The calibrated decay rate (Fig. 6) shows night T90 values of 4 d (equivalent to $k = 0.576 \, \text{d}^{-1} = 0.024 \, \text{h}^{-1}$) and peak daily values of 1.33 d (equivalent to $k = 1.73 \, \text{d}^{-1} = 0.072 \, \text{h}^{-1}$). De Marchis et al. (2013) and Scroccaro et al. (2010) used fixed day and night T90 values of 1 and 2 d for seawater. For river and lake waters (which are supposed to have slower decay rates compared to salty seawater) Jalliffier-Verne et al. (2016) used fixed day and night decay rates of 0.011 and $0.037 \, \text{h}^{-1}$, and Passerat et al. (2011) used $0.045 \, \text{h}^{-1}$. In this case the same fixed decay rates were applied to calibration events during both winter and summer periods even though winter decay rates are likely to be slower due to lower water temperature and solar radiation.

Figure 7a, c and e show the simulated versus the observed *E. coli* concentrations for the calibration and validation events at the near-shore sampling point at Pont del

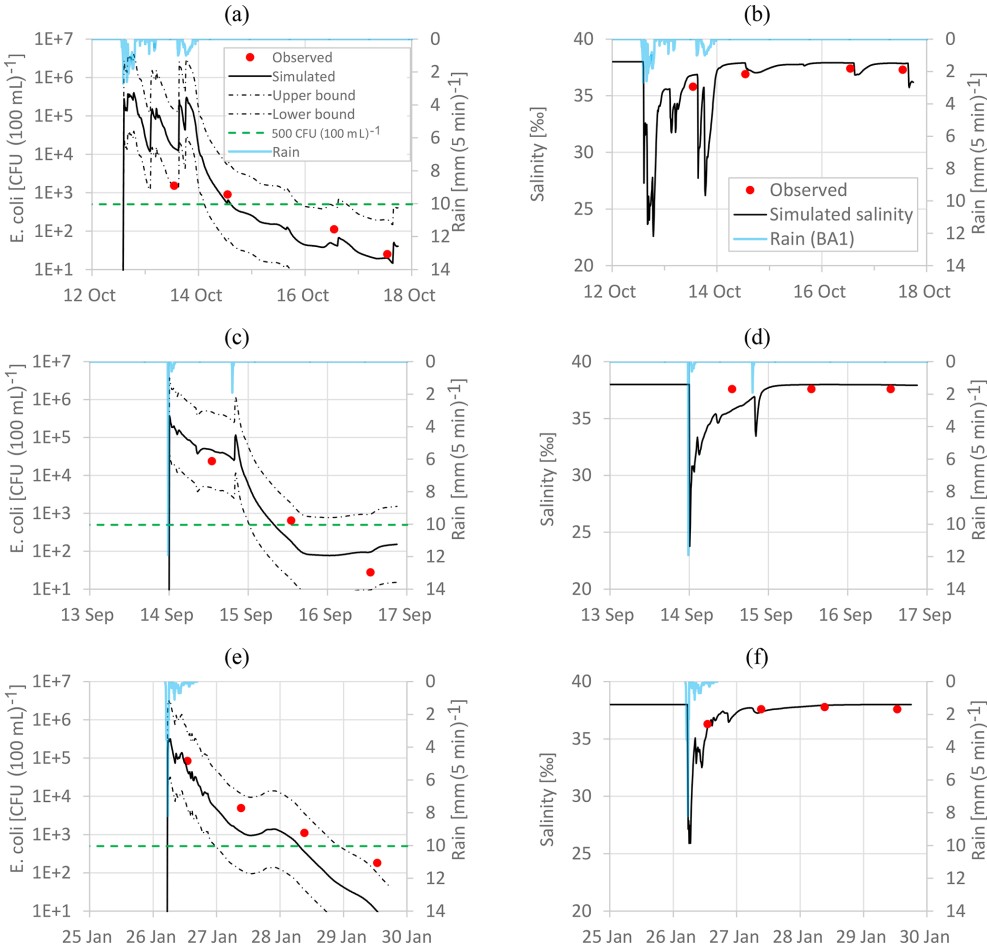

**Figure 7.** Near-shore simulated and observed *E. coli* concentrations **(a, c, e)** and salinity **(b, d, f)** for the calibration and validation events.

Petroli. Upper and lower bounds of the simulated concentrations are also shown and were obtained by running the model using both $10^7$ and $10^5$ CFU $(100\,\text{mL})^{-1}$ as fixed *E. coli* concentration inputs at the CSO points (the continuous black line uses $10^6$ CFU $(100\,\text{mL})^{-1}$). Despite the sparse data, the model seems to be able to reproduce the observed concentrations with an-order-of-magnitude precision. Figure 7b, d and f show the simulated versus the observed salinity concentrations at the near-shore sampling point at Pont del Petroli. It is noted that during rainfall events the nearshore seawater salinity falls significantly due to the discharge of non-salty CSO water into the sea of Badalona. The seawater salinity recovers to typical seawater concentrations (37.5‰–38‰) within some hours, faster than the time it takes for *E. coli* to recover below $500\,\text{CFU}\,(100\,\text{mL})^{-1}$. Due to the daily resolution of observed data, observed peaks of *E. coli* and salinity were poorly detected.

Figure 8 shows scatter plots of simulated and observed *E. coli* concentrations (panel a) and salinity (panel b). The simulated *E. coli* concentrations are shown to reproduce the observed ones with an-order-of-magnitude accuracy. *E.*

*coli* concentration accuracy of an order of magnitude is considered acceptable (Pongmala et al., 2015), particularly when simulating concentrations in receiving water bodies where models can be assessed at the order-of-magnitude level (Dorner et al., 2006). Salinity also seems to somehow follow the 1 : 1 line of simulated versus observed concentrations even though the majority of the observed values fall within 36‰ and 38‰, which might be a small range compared to the simulated values that can get down to less than 25‰ due to CSOs (Fig. 7b, d and f).

Table 3 shows the model performance parameters for the calibration and validation events. Log (base 10)-transformed root mean square error (LTRMSE), mean square log (with base 10) error (MSLE), and Pearson product moment correlation (PPMC) were used for *E. coli*, and root mean square error (RMSE) was used for salinity (Bennett et al., 2013; Hauduc et al., 2015). Few studies reported model performance errors for *E. coli* concentrations in receiving water bodies. Thupaki et al. (2010) obtained an LTRMSE of 0.41 and Liu et al. (2006) in the range of 0.705–0.835. Chen et

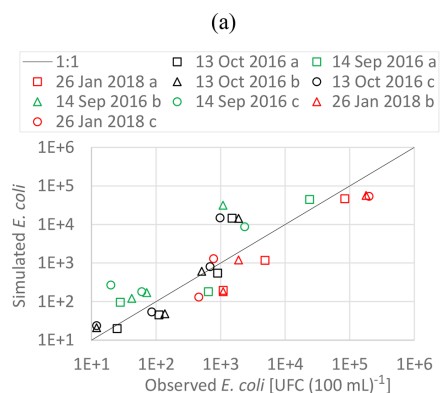 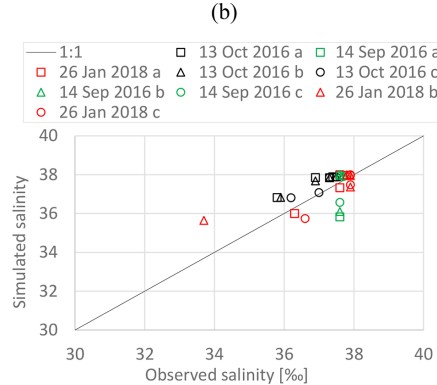

**Figure 8.** Scatter plots of simulated and observed *E. coli* concentrations **(a)** and salinity **(b)**. TS5

**Table 3.** Model performance parameters.

| Calibration and validation events | *E. coli* concentration | MSLE | 0.44 |
|---|---|---|---|
| | | LTRMSE | 0.66 |
| | | Pearson correlation coefficient | 0.83 |
| | Salinity concentration | RMSE (‰) | 0.73 |

**Table 4.** Hazard criteria based on *E. coli* concentration in seawater.

| Hazard criteria | *E. coli* concentration (CFU $(100\,\text{mL})^{-1}$) |
|---|---|
| Low | $< 250$ |
| Medium | $250 < x < 500$ |
| High | $> 500$ |

al. (2019) obtained RMSE values for salinity simulations of an estuary in the range of 0.13–3.01.

It is noted that the three events used for calibration and validation of the seawater quality model are from September, October and January. It could be relevant to look at more events during the bathing season that is the focus period of the hazard assessment.

### 2.5 Hazard assessment

Table 4 shows the hazard criteria applied. The selected approach is similar to the one proposed in the BWD and the Real Decreto 1341/2007 to classify bathing water quality. High hazard (*E. coli* $> 500\,\text{CFU}\,(100\,\text{mL})^{-1}$) is here considered as insufficient bathing water quality.

Hazard is quantified based on two novel indicators proposed that were computed for every beach of Badalona by continuously simulating nine consecutive bathing seasons (here assumed to be from 1 June to 1 September) from 2006 to 2014 using the coupled (urban drainage–seawater quality) model:

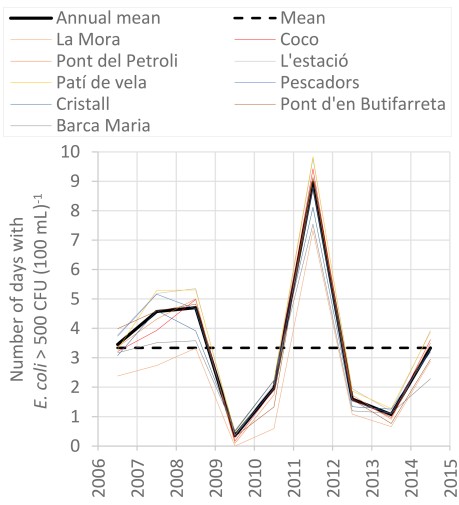

**Figure 9.** CE4 Simulated number of days per bathing season with high hazard (*E. coli* concentrations $> 500\,\text{CFU}\,(100\,\text{mL})^{-1}$) at all the different beaches of Badalona.

- the duration of insufficient bathing water quality over a bathing season;

- the duration of insufficient bathing water quality after CSO/rain events (in particular, the duration of insufficient bathing water quality is presented as a function of the event rainfall volume).

Both wind data and rainfall data between 2006 and 2014 (9 years) were obtained from the station of Fabra (approximately 10 km away from Badalona) located in the neighbor-

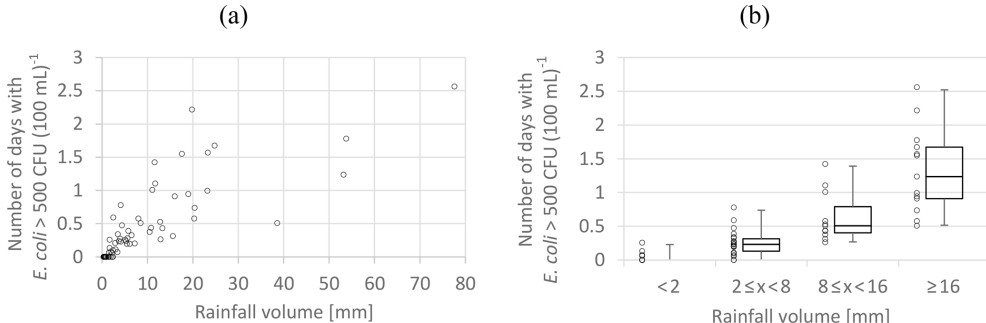

**Figure 10. (a)** Correlation between the simulated number of days with seawater *E. coli* concentrations above 500 CFU (100 mL)$^{-1}$ at Coco Beach and the rainfall volume. **(b)** The rainfall volume is categorized into four different ranges. The whisker boxes show 1st, 25th, 50th, 75th and 99th percentiles.

ing city of Barcelona. Data from Fabra were used as 9 years of continuous high-resolution data were not available from Badalona.

Hazard maps were also analyzed. However, together with the project stakeholders, they were not considered to be an interesting output because of the time and spatial variation of hazard during each different pollution event. Finally, a validation of the hazard assessment is shown for the duration of insufficient bathing water quality after CSO/rain events (Sect. 3).

## 3 Results

### 3.1 Hazard assessment

Figure 9 shows the average number of days with high hazard at the different beaches of Badalona and for the nine consecutive bathing seasons from 2006 to 2014. The annual average number of days (the thick black line) with high hazard is between 0 and 9 d per bathing season every year with an overall mean of 3–4 d per bathing season (the dashed black line). The results show a high variability that is highly related to the number and volume of rainfall events occurring during the different bathing seasons. The variability among the different beaches during the same bathing season is smaller compared to the variability due to different years.

Successively, the duration of every single seawater pollution event (defined in Fig. 2) was correlated to the total rainfall volume that fell during the same seawater pollution event (the total rain event duration of Fig. 2), and the results are shown in Fig. 10a. Figure 10a shows an example of the results from Coco Beach, even though all beaches were analyzed and all the graphs can be found in the delivery D4.4 of the BINGO project. Overall, the results show that the higher the rainfall volume, the longer the time period the beach is exposed to *E. coli* concentrations above threshold; however, above 15–25 mm of rain volume the increasing tendency seems to vanish, and only rainfall above a few millimeters can cause seawater *E. coli* concentrations

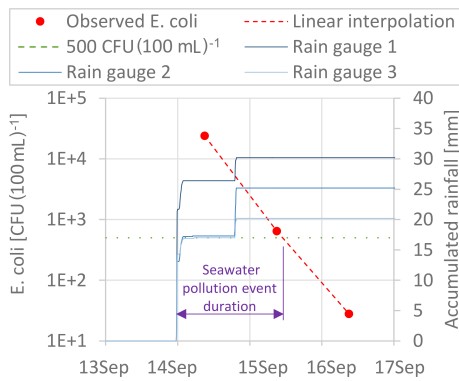

**Figure 11.** Example of how the duration of a seawater pollution event was graphically obtained.

> 500 CFU (100 mL)$^{-1}$. The large spreading of the correlation plots is mainly due to the different total rain event duration and the magnitude of marine currents. Overall, longer rain events produce longer CSOs and therefore longer seawater pollution events. Similarly, stronger winds and a rougher sea produce shorter seawater pollution events.

Figure 10b is a rearrangement of Fig. 10a and shows the probability distribution of seawater pollution events as a function of the rainfall volume. The discretization of the rainfall volume ($x$ axes) into four ranges was chosen in order to obtain both a reasonable number of events simulated in each range and volume ranges that are considered reasonable for local applications. This statistical approach that considered the total rainfall volume was considered the best one among several attempts of correlation between seawater pollution duration and rain intensities of different duration (e.g., 30, 60, 120 min rainfall).

Figure 10b provides one of the two main indicators proposed in this study: the duration of high hazard (insufficient bathing water quality) as a function of the event rainfall volume. For instance, an event of 12 mm rainfall (which would fall in the bin of $8 \le x < 16$ mm of Fig. 10b) is estimated to produce a median of 0.5 d of high hazard at Coco Beach.

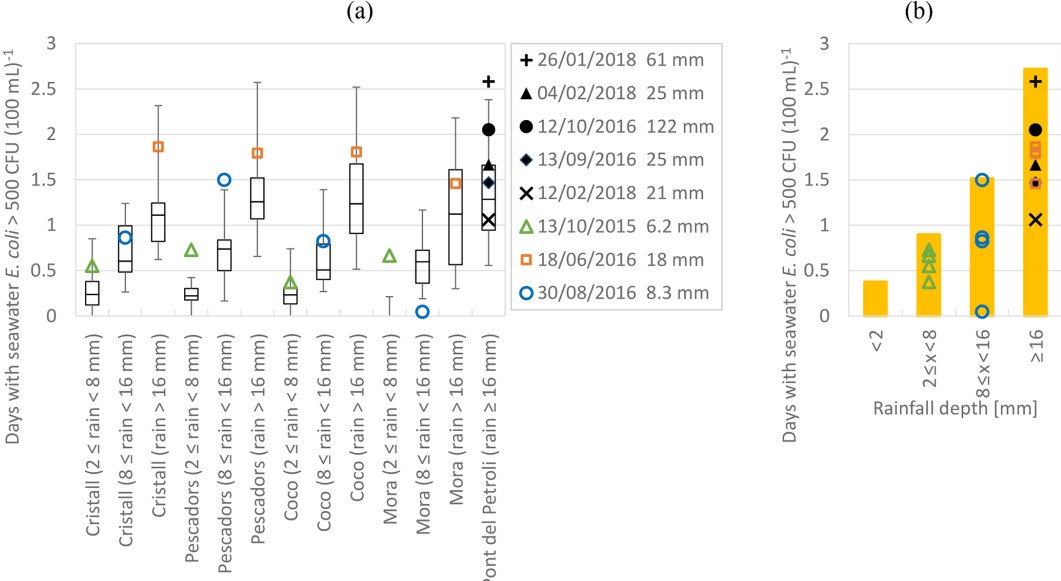

**Figure 12. (a)** Comparison between the estimated and the observed duration of high hazard (seawater *E. coli* > 500 CFU (100 mL)$^{-1}$). The whisker boxes show 1st, 25th, 50th, 75th and 99th percentiles of the duration of seawater pollution. **(b)** Maximum seawater pollution duration (orange bars) as a function of four different rainfall ranges.

The percentiles provided with the whisker boxes include an estimation of inter-event uncertainty obtained by continuous simulations using the deterministic coupled model. Other uncertainties like the ones associated with selected and calibrated parameters were not addressed.

**Validation of the hazard assessment**

A validation was performed only for the indicator of high hazard as a function of the event rainfall. The validation of the mean duration of high hazard per bathing season was not done due to lack of observed data. The number of days when bathing was forbidden during bathing seasons is available. However, these data cannot be compared with the simulated high hazard because bathing-forbidden days are dependent on local protocols that, for instance in the case of Badalona, allow the reestablishment of bathing permissions only after positive bathing water quality measurements which usually take more than 24 h to obtain.

Figure 11 shows an example of how the duration of a seawater pollution event was graphically obtained based on rainfall data and observed *E. coli* concentrations. The seawater pollution event is assumed to start when the accumulated rainfall exceeds 1 mm. An analysis of the simulation results showed that a seawater pollution event can start up to an hour later compared to the proposed beginning point. This depends on how far from the CSO the control point is and also on the CSO events which can start with a delay compared to the rainfall. The seawater pollution event is supposed to end when the simple linear interpolation (the dashed red line of Fig. 11) between measured concentrations

crosses the selected threshold of 500 CFU (100 mL)$^{-1}$. The total rainfall associated with the seawater pollution event is the average total volume (from the available rain gauges) that fell during the pollution event.

Figure 12a shows the comparison between the simulated and observed high hazard duration. Eight events were used for the comparison. Overall, the majority of the observed durations fall within the simulated 1st and 99th percentiles. However, there are some outliers. The two outliers at Mora Beach are likely because this beach is close to the mouth of the Besòs river, which might not be properly represented in the model. Also, there are several model uncertainties that were not simulated (for example, input parameters and calibration uncertainties). Further, it seems that the observed values are in the higher range of the simulated percentiles; this can be because all the CSO events that caused little seawater pollution could not be measured by the available sampling resolutions (approximately a sample per day). Overall, this can be considered as a preliminary visual validation.

For risk assessment purposes (as part of the BINGO project), together with the project stakeholders, it was decided to adjust/calibrate the proposed percentile duration in order to obtain the deterministic maximum durations of high hazard that are shown in Fig. 12b. For this purpose, several steps were applied: the observed seawater pollution duration derived from intestinal enterococci observations was compared to the simulated *E. coli* percentile durations of Fig. 12a; all the beaches were merged into a unique representative value of pollution duration obtained from the worst 99.9th percentile among all the beaches; finally, a further safety factor of 5 % was applied so that all the outliers would be ac-

commodated within the newly developed bars, representing a practical deterministic value of maximum seawater pollution duration as a function of four different rainfall ranges.

## 4    Conclusions

This study quantified the health hazard of bathing waters affected by CSOs based on two novel indicators: the mean duration of insufficient bathing water quality (1) per bathing season and (2) after single CSO/rain events. Overall, a great uncertainty is associated with the evaluated pollutant hazard, mainly due to the variability of water quality variables, rainfall patterns and seawater currents. A novel correlation between the duration of seawater pollution and the event rainfall volume was presented. Also, a coupled urban drainage and seawater quality model was developed, calibrated and validated based on local observations. Furthermore, hazard assessment was based on a statistical analysis of the continuous simulation results of nine consecutive bathing seasons using the coupled model. Finally, a validation of the estimated hazard was also shown.

The pollutant hazard of bathing waters affected by CSOs was assessed for the case study of Badalona (Spain) even though the methodology presented can be considered generally applicable to other urban areas and related receiving bathing water bodies. The results of this study were useful as inputs for risk assessment and to analyze direct, indirect, tangible, and intangible impacts related to CSO events and consequent pollution of seawater. Also, the correlation presented to predict the duration of insufficient bathing water quality as a function of the observed rainfall volume can be useful to bathing water managers.

*Code and data availability.* Model files and data are not provided due to the confidentiality of the data and models shared among the local project stakeholders of Badalona. Notwithstanding, in agreement with the other project local stakeholders, the authors of this paper will try to address specific requests for scientific purposes.

*Author contributions.* BR, MM and LL coordinated the research project. AAO and JCSC managed the data collection. LL, BR and EMG developed the conceptual model, and LL set up the model, wrote the code and performed the simulations. LL prepared the paper with the contributions of all co-authors.

*Competing interests.* The authors declare that they have no conflict of interest.

*Special issue statement.* This article is part of the special issue "Integrated assessment of climate change impacts at selected European

research sites – from climate and hydrological hazards to risk analysis and measures". It is not associated with a conference.

*Acknowledgements.* This study was conducted as part of the BINGO European H2020 project (http://www.projectbingo.eu/, last access: 1 April 2020). The authors thank the Municipality of Badalona and particularly Antonio Gerez Angulo, Maria Luisa Forcadell Berenguer, Gregori Muñoz-Ramos Trayter and Josep Anton Montes Carretero for their valuable contributions. Also, the authors wish to thank the LIFE EFFIDRAIN project (LIFE14 ENV/ES/000860, http://www.life-effidrain.eu/, last access: 1 April 2020) for sharing data.

*Financial support.* This research has been supported by the BINGO European H2020 project (grant no. 641739).

*Review statement.* This paper was edited by Adriana Bruggeman and reviewed by Ekaterina Sokolova and Vasilis Bellos.

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

**Remarks from the language copy-editor**

CE1  The article here is grammatically correct.

CE2  Please verify the rephrased sentence.

CE3  I inserted your changes, but this list still requires the use of semicolons as the items include internal punctuation. Please make sure all the items have been grouped correctly or clarify.

CE4  We inserted the unit changes you requested and made the units consistent throughout the figures. Please check all the figure contents carefully.

**Remarks from the typesetter**

TS1  Please note that it is our standard to abbreviate SI-accepted units in combination with numbers, including d for "day".

TS2  According to our standards, all changes in values must first be approved by the editor, as data have already been reviewed, discussed and approved. In case these changes are necessary, the explanation you provided and the requested changes will be forwarded to the editor. Please note that this entire process will be available online after publication. Upon approval, we will make the appropriate changes. Thank you for your understanding.

TS3  According to our standards, all changes in values must first be approved by the editor, as data have already been reviewed, discussed and approved. In case these changes are necessary, the explanation you provided and the requested changes will be forwarded to the editor. Please note that this entire process will be available online after publication. Upon approval, we will make the appropriate changes. Thank you for your understanding.

TS4  According to our standards, a skinny space is only used for values with five digits or more.

TS5  Please check x-axis of panel a: should it be "UFC" or "CFU"?.

TS6  Please provide date of last access.

TS7  If the refenrece is not available under the provided link, please provide the place of publication/publisher's location.