# Peer review of "Modelling of E. coli distribution for hazard assessment of bathing waters affected by combined sewer overflows"

_Natural Hazards and Earth System Sciences, 2019_

## Referee Comment (RC1) · Ekaterina Sokolova (Referee) · 5 Nov 2019

General comments (GC)

This manuscript reports on a how a modelling study was used to obtain a relation between the duration of seawater pollution events and rainfall volume. Two models were used: an urban drainage model to simulate combined sewer overflows (CSOs) and a hydrodynamic model to simulate the spread of pollution in the nearshore area. This work contributes to the existing body of literature on the impact of CSOs on the water quality. In my opinion, the scientific significance and scientific quality are good, while the presentation of the manuscript can be improved.

GC1

[Figure]

Consider restructuring the manuscript to improve clarity and flow – see the proposed outline below. The suggestion is to report the calibration and validation of each of the two models directly next to the description of the modelling set-up, to make it easier for the reader to follow.

Proposed outline:

1 Introduction

2 Methods

2.1 Study area

2.2 Hazard assessment (the hazard levels were defined; the coupled model was used to obtain the data for hazard assessment)

2.3 Urban drainage model

2.3.1 Model set-up of the urban drainage model (Model type and software; Input data: types of data and sources of these data, assumptions)

2.3.2 Calibration and validation of the urban drainage model (Data used for calibration and validation (what periods and why); Calibrated parameters (which and why, selected values); Calibration and validation results, e.g. graphs, RMSE, etc.)

2.4 Hydrodynamic model

2.4.1 Model set-up of the hydrodynamic model (Model type and software; Input data: types of data and sources of these data, assumptions)

2.4.2 Calibration and validation of the hydrodynamic model (Data used for calibration and validation (what periods and why); Calibrated parameters (which and why, selected values); Calibration and validation results, e.g. graphs, RMSE, etc.)

3 Results and Discussion

3.1 Hazard assessment

3.2 Validation of hazard assessment

3.3 Applicability of the suggested approach (here, the limitations and advantages of the modelling approaches and of the hazard assessment approach can be discussed, also in the context of existing literature)

4 Conclusions

GC2

Consider revising which tables and figures are necessary to include in the manuscript, which can be placed as supplementary material, and which can be omitted. The quality of the figures can be improved to make them clearer and more informative.

GC3

Consider more clearly stating which data and which periods/events were used for calibration and validation of the different models and methods, with motivation why. Consider showing more (all?) graphs for visual comparison of the modelled and observed data, either in the main text (if appropriate) or as supplementary material.

GC4

Consider including a (more detailed) discussion of the applicability and limitations of the used modelling approaches and of the developed hazard assessment approach, also in the context of other studies.

Specific comments (SC)

SC1

Title: I would suggest mentioning in the title that the modelling approach was used. Also, I would recommend against using the word "health" in the title, because health or infection risks are beyond the scope of the paper, the paper is about E. coli levels representing the faecal pollution of the water.

SC2

Line 52-53: "None of the studies presented above provided a methodology to evaluate health hazard of bathing waters affected by CSOs that is the main aim of this study" I would suggest reformulating this statement. There are examples in the literature combining bathing water quality modelling with quantitative microbial risk assessment, i.e. to evaluate health hazards.

SC3

Consider explaining what BINGO is (very briefly).

SC4

Line 90 and onward: Automatic samplers – how often/how many times/when were they used? Why is it relevant to this study? See also my suggestion to restructure the presentation of the data used in the study (GC1).

SC5

Line 97: Consider specifying how long the pedestrian bridge is – here or where appropriate.

SC6

Section 2.3: The data – It could be better to mention the different types or data where relevant instead, for example, in the sections about the urban drainage model, in the section on hydrodynamic model, in the section of hazard assessment. See also GC1.

SC7

Line 105-110: The description of what is shown in Figure 3 is unclear, consider revising. For example, it is written that there are 4 measurements in Figure 3a while there are 2 measurements in Figure 3b – unclear how this is meant, there are many points in these figures. Also, Figure 3b indicates two locations (R.C. M.A.). Consider improving

the text and Figure 3.

SC8

Section 2.4.1: Consider clearly stating which model was used to simulate the urban drainage processes. It is mentioned that the original model (using which software?) was integrated into InfoWorks (consider explaining what InfoWorks is – it is not very clear from the provided website). Then it is stated that runoff-rainfall was simulated using SWMM. What model was used for simulating the flow in the sewers? Also, see comment GC1.

SC9

Line 185: Figure 1 indicated that there are several rain gauges in the city of Badalona. On line 185 it is mentioned that the rain data were used from the city of Barcelona – why? Consider motivating why local data were not used. The same is relevant for the wind data.

SC10

Table 1 is unnecessary, since this hazard classification is explained in the text.

SC11

Table 2: specify the unit for the return period.

SC12

Section 3.1: How were the calibration and validation events selected? Were data from other overflow events available? Why were the SCOs in 2017 not included in the calibration/validation procedure? See also GC3.

SC13

Section 3.1: Which parameters were calibrated?

SC14

Line 220: Why were these periods selected for calibration/validation? Were there data available for more events? Could it be more relevant to look at more periods during the bathing season? See also GC3.

SC15

Figure 6 can be placed in supporting information.

SC16

Figures 7 and 8: Consider showing the graphs for all three calibration/validation events. The results for E. coli (three graphs) and salinity (three graphs) could be combined into one figure with six graphs in total. This would provide more information on how the E. coli concentrations and salinity change during overflow events.

SC17

Table 4 is unnecessary since it presents only four values – it would be better to present these values in the text.

SC18

Consider improving Figure 10 to make it more easily readable.

SC19

Figure 12 can be placed in supporting information instead.

SC20

Line 317 "The validation of the mean duration of high hazard per bathing season was not done due to lack of observed data." Are not data available on how many days the beaches were closed during each bathing season?

SC21

Line 330 It is not very clear what is meant here about the percentiles – consider

rephrasing.

SC22

In general, I think it would be good to include a table that summarises which events (periods of time) and types of data were used for calibration and validation of the models and approaches: sewer model, hydrodynamic model, hazard assessment. I think this should be explained where relevant in the text (in separate sections where the models are described), but a summarising table can be provided as supporting material.

SC23

In general, in the figures, make sure that it is clear whether the figure shows observed (measured data) or modelling results (see e.g. Figure 11).

SC24

Section 3.3.1 and conclusions: Was the purpose of developing this approach to predict the duration of water pollution events based on the rainfall volume? What are the practical implications of this work? Can this method be used by water managers? Any other reflections about the significance of the findings?

SC25

Figure 13: How were the events for validation of this approach selected? Were there more data available? Currently, measurements/estimations for three years are presented, with some of the measurements being outside of the bathing season. See also GC3.

SC26

Conclusions – first line: I think it is better not to call this "health" hazard, because the health and health risks (measured in e.g. probability of infection or DALYs) were not calculated – beyond the scope of this study. See also SC1.
SC27

Conclusions: "A novel correlation between the duration of sea water contamination and the event rainfall volume was presented." Consider discussing (in the appropriate section of the text) whether there are other studies attempting to do something similar – correlate precipitation, impact of CSOs with bathing water quality.

Technical corrections

E. coli and Enterococcus intestinalis: small letter for the species name, Italics for Latin. The way E. coli is written needs to be corrected everywhere in the manuscript, including figures.

My impression is that "wastewater" is most often written as one word.

"Pollution" and "contamination" seem to be used interchangeably in the manuscript. Consider if it would be better to use one term only, if no difference is meant.

Please also note the supplement to this comment:
https://www.nat-hazards-earth-syst-sci-discuss.net/nhess-2019-292/nhess-2019-292-RC1-supplement.pdf

———————————————

---

## Referee Comment (RC2) · Vasilis Bellos (Referee) · 4 Dec 2019

The manuscript titled "Evaluating health hazard of bathing waters affected by combined sewer overflows" introduces the use of new health hazard indicators for bathing water, correlating the rainfall volume with the duration of the bathing water contamination. These indicators are applied in a real world application.

For this reason, the authors coupled an urban drainage model using the Infoworks software with a sea water quality model using the MOHID software. The manuscript is well-written, well organized, well structured and is scientifically consistent. The authors provided all the required information and assumptions and did not hide the "weak" approximations. Their method is practical (and still consistent) and can be applied

directly to other case studies. I suggest to be published after some minor revisions.

My remarks:

1) In Table 2 there is the variable T in the brackets and I suppose is the return period calculated using the available IDF curves and rainfall duration. Regarding the rainfall intensity, I am a little bit surprised to see such small values (from less than year to 6 years). The authors shall double check these values.

2) Which is the way for calibrating the urban drainage model? The authors automatized Infoworks and used an optimization algorithm and if yes, which is the algorithm.

3) Which is the objective function for the calibration of the urban drainage model? The sum of the RMSEs?

4) The authors obtained a rather small value for the Manning coefficient at impervious areas (0.013 s/mˆ1.3). Although these areas are mad from asphalt or concrete and are characterized by small roughness, however in real world there are several obstacles in surface, increasing the roughness. Did the authors use constraints for keeping a physical meaning at the variables which were calibrated, or the parameters are considered as black-box parameters?

5) Except of the mentioned parameters at the urban drainage model which were calibrated, what did the authors with the rest, such as parameters for the hydraulic structure of the CSO? These are the parameters which were manually calibrated?

6) Regarding the sea water quality calibration the authors might provide a magnitude of the computational time (hours, days?) in order to support their decision for manual calibration. Besides they could provide some additional references about the use of Machine Learning in such cases, when a highly computationally demanding software requires calibration. Finally, taking into account the nature of the observations (some points vs. a dynamic time-series), the process followed by the authors is more a plausible check than a proper calibration and they should discuss about that.

7) The authors classified the rainfall volume to bins (e.g. Fig 11) but the range between 8-10 mm is not appearing. Is there a reason for that?

8) Can the authors support bibliographically their decision to assume that the evolution of E. Coli concentration is done with this linear way or it is an assumption for practical reasons (this does not reduce the importance of their work).

---

## Author Response (AR1)

First of all, the authors thank the Editor and the reviewers for their constructive and valuable feedbacks.

Please find the authors' replies to the comments of the Editor written in blue.

**EDITOR**

Comments to the Author:

The research work is impressive and useful. The presentation of the work needs improvement to make the knowledge accessible and citeable. The reviews were helpful and it looks like the authors made many improvements. However, from the authors' response to the reviewers' comments, it is often impossible to see what changes you made. You need to indicate in your response how you changed it. You can copy-paste your corrections from the manuscript in the response or you can indicate the line numbers of the corrections and add a manuscript with track changes. I have a few additional comments that need to be addressed.

**GENERAL COMMENTS**

(1) Add a section in the Methods Section that explains the calibration, validation and model application (model parameters, objective functions, events). Specify the time interval of the data used for RMSE. Remove methods from the Results Section.

The calibration and validation section of both the urban drainage and the sea water quality model was moved into the methodology part. Also, model parameters, objective functions, time interval of the data used for RMSE were specified. (From line 230 to 386 of the revised manuscript with changes marked)

(2) Describe the selection of model input data (e.g., use of data from Figure 3 and Literature, e.g., l.224-238) in a section in the Methods.

The calibration and validation sections that included the description of both fixed and calibrated parameters were moved to the methodology section along with different clarifications requested by the specific comments provided by the reviewers. (From line 230 to 258 and lines 310 to 317 of the revised manuscript with changes marked)

(3) Deliverables of the BINGO project are grey literature. Try to limit references to grey literature as much as possible. Add Supplementary information instead, as also requested by RC1 (GC2).

We removed some BINGO references and added several graphs to the manuscript (4 graphs were added to Figure 7). Nevertheless, there are many calibration and validation graphs both from the urban drainage and the sea water quality model. We like to leave all the remaining graphs as referenced the 2 public deliverables of BINGO (we have specified the URL to the public pdf): D3.3 and D4.1.

(4) Put reference to the BINGO project in the acknowledgements, not in the scientific manuscript.

We removed the general BINGO reference and add the project URL in the acknowledgements.

(5) I would strongly suggest to add you data sets in ZENODO and add a reference to these data sets (you will be the author).

Due to the confidentiality of the data shared among the local project stakeholders of Badalona, the authors prefer to avoid the publication of the dataset on ZENODO. Notwithstanding, other authors can contact us and, in agree with other project local stakeholders, we will try to provide specific data needed for specific scientific purposes.

SPECIFIC COMMENTS

l.88-93: Specify when the sensors and samplers were installed and operational.

Done

l.99: potential oxygen reduction - please explain how this is measured.

In order to avoid confusion we have removed this indicator as it was not used in this study.

l.122-123: The model simulates rainfall-runoff processes, domestic and other types of sewage water produced …

Specify that only the rainfall-runoff processes are used for this application.

Actually all the processes were included into the simulation of urban water quantity. We clarified this point. (line 259 of the revised manuscripts with changes marked)

l.175-176: Don't repeat the table in the text, either use a table or use text. (See also RC1, SC10).

Ok. We left the table and removed the 2 lines of text.

l.180: simulating

Ok.

l.213-217: Manual calibration also belongs in the Methods. Specify what is calibrated (which parameters are changed) and why would this not affect the results. You want to make sure that it "did not affect" instead of "would not affect".

Ok.

I am looking forward to receiving an improved response to the comments of the reviewers and to the above comments!

**REVIEWER 1**

General comments (GC)

This manuscript reports on a how a modelling study was used to obtain a relation between the duration of seawater pollution events and rainfall volume. Two models were used: an urban drainage model to simulate combined sewer overflows (CSOs) and a hydrodynamic model to simulate the spread of pollution in the nearshore area. This work contributes to the existing body of literature on the impact of CSOs on the water quality. In my opinion, the scientific significance and scientific quality are good, while the presentation of the manuscript can be improved.

GC1

Consider restructuring the manuscript to improve clarity and flow – see the proposed outline below. The suggestion is to report the calibration and validation of each of the two models directly next to the description of the modelling set-up, to make it easier for the reader to follow.

Proposed outline:

1 Introduction

2 Methods

2.1 Study area

2.2 Hazard assessment (the hazard levels were defined; the coupled model was used to obtain the data for hazard assessment)

2.3 Urban drainage model

2.3.1 Model set-up of the urban drainage model (Model type and software; Input data: types of data and sources of these data, assumptions)

2.3.2 Calibration and validation of the urban drainage model (Data used for calibration and validation (what periods and why); Calibrated parameters (which and why, selected values); Calibration and validation results, e.g. graphs, RMSE, etc.)

2.4 Hydrodynamic model

2.4.1 Model set-up of the hydrodynamic model (Model type and software; Input data: types of data and sources of these data, assumptions)

2.4.2 Calibration and validation of the hydrodynamic model (Data used for calibration and validation (what periods and why); Calibrated parameters (which and why, selected values); Calibration and validation results, e.g. graphs, RMSE, etc.)

3 Results and Discussion

3.1 Hazard assessment

3.2 Validation of hazard assessment

3.3 Applicability of the suggested approach (here, the limitations and advantages of the modelling approaches and of the hazard assessment approach can be discussed, also in the context of existing literature)

4 Conclusions

Reviewer 1 proposes a restructuring of the manuscript in order to improve clarity and flow, whereas Reviewer 2 finds the paper "well organized and well structured".

In order to improve clarity and flow we have significantly improved the old Section 2.3 and changed the title of the old section 2.2 following the reviewer's suggestions relative also to the Specific Comments SC4, SC6 and SC8.

We moved the calibration and validation sections to the methodology as requested. The second suggestion of Reviewer 1 is to remove the data section 2.3 (a section dedicated to all the data used in the whole study) and describe the data separately in each section ('urban drainage model calibration and validation', 'sea water quality model calibration and validation' and 'Validation of the hazard assessment'). We believe it is better to keep the data section separately in order to avoid many repetitions that would occur. This is because the same data are often used in different sections: for instance some of the data used in the section 'sea water quality model calibration and validation' are the same data used in the section 'Validation of the hazard assessment'. The data section was improved by adding several lines guiding the reader to improve readability.

GC2

Consider revising which tables and figures are necessary to include in the manuscript, which can be placed as supplementary material, and which can be omitted. The quality of the figures can be improved to make them clearer and more informative.

Figures 7 and 8 were merged into one figure and 3 more graphs were added. Figure 10 was modified in order to make it clearer. Figure 3 was corrected. 2 out of 4 graphs were removed from Figure 11. We did not consider adding supplementary material as all the additional figure can be found in the public deliverables of the BINGO project.

GC3

Consider more clearly stating which data and which periods/events were used for calibration and validation of the different models and methods, with motivation why. Consider showing more (all?) graphs for visual comparison of the modelled and observed data, either in the main text (if appropriate) or as supplementary material.

We have added some sentences stating which data and which periods/events were used for calibration and validation of the different models and methods (Line 231-240 of the revised manuscript with changes marked). Also, we have added more graphs for visual comparison of the modelled and observed data. (Figure 7 of the revised manuscript with changes marked)

Consider including a (more detailed) discussion of the applicability and limitations of the used modelling approaches and of the developed hazard assessment approach, also in the context of other studies.

We consider that the applicability and limitations of the used modeling approaches were already sufficiently discussed throughout the paper also with comparison to other studies. We have not added more details as suggested by the reviewer.

Specific comments (SC)

SC1

Title: I would suggest mentioning in the title that the modelling approach was used. Also, I would recommend against using the word "health" in the title, because health or infection risks are beyond the scope of the paper, the paper is about E. coli levels representing the fecal pollution of the water.

We agree with the reviewer and we have modified the title.

SC2

Line 52-53: "None of the studies presented above provided a methodology to evaluate health hazard of bathing waters affected by CSOs that is the main aim of this study" I would suggest reformulating this statement. There are examples in the literature combining bathing water quality modelling with quantitative microbial risk assessment, i.e. to evaluate health hazards.

We have improved this sentence and part of the following ones. (Lines 57-68 of the revised manuscript with changes marked)

SC3

Consider explaining what BINGO is (very briefly).

We have added a line at the end of the abstract.

SC4

Line 90 and onward: Automatic samplers – how often/how many times/when were they used? Why is it relevant to this study? See also my suggestion to restructure the presentation of the data used in the study (GC1).

We added some lines in order to clarify these points. (Lines 105-111 of the revised manuscript with changes marked)

SC5

Line 97: Consider specifying how long the pedestrian bridge is – here or where appropriate.

Ok. (Line 140 of the revised manuscript with changes marked)

SC6

Section 2.3: The data – It could be better to mention the different types or data where relevant instead, for example, in the sections about the urban drainage model, in the section on hydrodynamic model, in the section of hazard assessment. See also GC1.

We believe that a unique section including all the data would be clearer. Nevertheless, we have improved the readability of the data section also following the suggestions of SC4.

SC7

Line 105-110: The description of what is shown in Figure 3 is unclear, consider revising. For example, it is written that there are 4 measurements in Figure 3a while there are 2 measurements in Figure 3b – unclear how this is meant, there are many points in these figures. Also, Figure 3b indicates two locations (R.C. M.A.). Consider improving the text and Figure 3.

We clarified the text and corrected the figure. (Lines 122-134 of the revised manuscript with changes marked)

SC8

Section 2.4.1: Consider clearly stating which model was used to simulate the urban drainage processes. It is mentioned that the original model (using which software?) was integrated into InfoWorks (consider explaining what InfoWorks is – it is not very clear from the provided website). Then it is stated that runoff-rainfall was simulated using SWMM. What model was used for simulating the flow in the sewers? Also, see comment GC1.

We clarified these points. (Lines 161-168 of the revised manuscript with changes marked)

SC9

Line 185: Figure 1 indicated that there are several rain gauges in the city of Badalona. On line 185 it is mentioned that the rain data were used from the city of Barcelona – why? Consider motivating why local data were not used. The same is relevant for the wind data.

We explained this. (Lines 397-400 of the revised manuscript with changes marked)

SC10

Table 1 is unnecessary, since this hazard classification is explained in the text.

We prefer to keep this table as part of the manuscript as it shows the hazard criteria that are a main input for the methodology.

SC11

Table 2: specify the unit for the return period.

Ok.

SC12

Section 3.1: How were the calibration and validation events selected? Were data from other overflow events available? Why were the SCOs in 2017 not included in the calibration/validation procedure? See also GC3.

We clarified this and improved the description of the section. (Lines 231-239 of the revised manuscript with changes marked)

SC13

Section 3.1: Which parameters were calibrated?

We clarified this. (Lines 249-256 of the revised manuscript with changes marked)

SC14

Line 220: Why were these periods selected for calibration/validation? Were there data available for more events? Could it be more relevant to look at more periods during the bathing season? See also GC3.

We clarified this and added a line at the end of the Section. (Lines 310-311 and 384-386 of the revised manuscript with changes marked)

SC15

Figure 6 can be placed in supporting information.

We prefer to keep this figure as part of the manuscript as it shows the temporal variation of one of the most influential and calibrated model parameters.

SC16

Figures 7 and 8: Consider showing the graphs for all three calibration/validation events. The results for E. coli (three graphs) and salinity (three graphs) could be combined into one figure with six graphs in total. This would provide more information on how the E. coli concentrations and salinity change during overflow events.

Figures 7 and 8 were merged into one figure of six graphs as recommended by the reviewer.

SC17

Table 4 is unnecessary since it presents only four values – it would be better to present these values in the text.

We prefer to keep the table in the manuscript as we consider that it helps readability.

SC18

Consider improving Figure 10 to make it more easily readable.

We modified Figure 10.

SC19

Figure 12 can be placed in supporting information instead.

We prefer to keep this figure as part of the manuscript as it is a visual example that helps the reader understanding the procedure adopted.

SC20

Line 317 "The validation of the mean duration of high hazard per bathing season was not done due to lack of observed data." Are not data available on how many days the beaches were closed during each bathing season?

We clarified this. (Lines 443-447 of the revised manuscript with changes marked)

SC21

Line 330 It is not very clear what is meant here about the percentiles – consider rephrasing.

We clarified this. (Lines 435 of the revised manuscript with changes marked)

SC22

In general, I think it would be good to include a table that summarises which events (periods of time) and types of data were used for calibration and validation of the models and approaches: sewer model, hydrodynamic model, hazard assessment. I think this should be explained where relevant in the text (in separate sections where the models are described), but a summarising table can be provided as supporting material.

A summary table was not introduced as it is was not considered to be relevant since the text was improved and all the suggested details related to the calibration and validation processes are now present in the text.

SC23

In general, in the figures, make sure that it is clear whether the figure shows observed (measured) data) or modelling results (see e.g. Figure 11).

The caption of Figure 11 was modified.

SC24

Section 3.3.1 and conclusions: Was the purpose of developing this approach to predict the duration of water pollution events based on the rainfall volume? What are the practical implications of this work? Can this method be used by water managers? Any other reflections about the significance of the findings?

These questions were already addressed in the text. Further reflections about the significance of the findings were not added.

SC25

Figure 13: How were the events for validation of this approach selected? Were there more data available? Currently, measurements/estimations for three years are presented, with some of the measurements being outside of the bathing season. See also GC3.

We added some lines in the data section in order to clarify this.

SC26

Conclusions – first line: I think it is better not to call this "health" hazard, because the health and health risks (measured in e.g. probability of infection or DALYs) were not calculated – beyond the scope of this study. See also SC1.

We like to keep "health hazard" as also the Bathing Water DIRECTIVE 2006/7/EC treats the bathing water quality as a hazard for human health.

SC27

Conclusions: "A novel correlation between the duration of sea water contamination and the event rainfall volume was presented." Consider discussing (in the appropriate section of the text) whether there are other studies attempting to do something similar – correlate precipitation, impact of CSOs with bathing water quality.

To our knowledge there were no other studies attempting something similar.

Technical corrections

E. coli and Enterococcus intestinalis: small letter for the species name, Italics for Latin. The way E. coli is written needs to be corrected everywhere in the manuscript, including figures.

Text and figures were corrected.

My impression is that "wastewater" is most often written as one word.

Wastewater was modified throughout the manuscript.

"Pollution" and "contamination" seem to be used interchangeably in the manuscript. Consider if it would be better to use one term only, if no difference is meant.

Ok. We used pollution which seems to be more appropriate for our case according to the following definitions: http://www.fao.org/3/x5624e/x5624e04.htm#1.1%20contamination%20or%20pollution.

REVIEWER 2

The manuscript titled "Evaluating health hazard of bathing waters affected by combined sewer overflows" introduces the use of new health hazard indicators for bathing water, correlating the rainfall volume with the duration of the bathing water contamination.
These indicators are applied in a real world application. For this reason, the authors coupled an urban drainage model using the Infoworks software with a sea water quality model using the MOHID software. The manuscript is well-written, well organized, well structured and is scientifically consistent. The authors provided all the required information and assumptions and did not hide the "weak" approximations. Their method is practical (and still consistent) and can be applied directly to other case studies. I suggest to be published after some minor revisions.

My remarks:
1) In Table 2 there is the variable T in the brackets and I suppose is the return period calculated using the available IDF curves and rainfall duration. Regarding the rainfall intensity, I am a little bit surprised to see such small values (from less than year to 6 years). The authors shall double check these values.

We clarified how the return period was calculated and checked all and corrected some of the values. (Table 2 of the revised manuscript with changes marked)

2) Which is the way for calibrating the urban drainage model? The authors automatized Infoworks and used an optimization algorithm and if yes, which is the algorithm.

We clarified this. (Line 231 of the revised manuscript with changes marked)

3) Which is the objective function for the calibration of the urban drainage model? The sum of the RMSEs?

We clarified this. (Line 231 of the revised manuscript with changes marked)

4) The authors obtained a rather small value for the Manning coefficient at impervious areas (0.013 s/m^1.3). Although these areas are mad from asphalt or concrete and are characterized by small roughness, however in real world there are several obstacles in surface, increasing the roughness. Did the authors use constraints for keeping a physical meaning at the variables which were calibrated, or the parameters are considered as black-box parameters?

We added some lines comparing literature values of the Manning coefficient with the ones obtained in our calibration. (Line 249-256 of the revised manuscript with changes marked)

5) Except of the mentioned parameters at the urban drainage model which were calibrated, what did the authors with the rest, such as parameters for the hydraulic structure of the CSO? These are the parameters which were manually calibrated?

We clarified this. (Line 266-272 of the revised manuscript with changes marked)

6) Regarding the sea water quality calibration the authors might provide a magnitude of the computational time (hours, days?) in order to support their decision for manual calibration. Besides they could provide some additional references about the use of Machine Learning in such cases, when a highly computationally demanding software requires calibration.

Finally, taking into account the nature of the observations (some points vs. a dynamic time-series), the process followed by the authors is more a plausible check than a proper calibration and they should discuss about that.

We added the computational time required. We did not add references about machine learning as we consider it beyond the scope of the paper. Finally, we have specified that sparse data and few events are commonly used for calibration and validation of similar studies (Some literature studies were also provided). (Line 309-317 of the revised manuscript with changes marked)

7) The authors classified the rainfall volume to bins (e.g. Fig 11) but the range between 8-10 mm is not appearing. Is there a reason for that?

The manuscript already includes an explanation for the selected rainfall ranges. We further clarified this in the manuscript. (Line 426-437 of the revised manuscript with changes marked)

8) Can the authors support bibliographically their decision to assume that the evolution of E. Coli concentration is done with this linear way or it is an assumption for practical reasons (this does not reduce the importance of their work).

We assume that the reviewer is talking about the E. Coli degradation rate. The manuscript already include some lines discussing the selected E. Coli degradation rates and comparison with literature (section "
[revised manuscript text omitted]